# Why Is My Classifier Discriminatory?

**Irene Y. Chen**
MIT
iychen@mit.edu

**Fredrik D. Johansson**
MIT
fredrikj@mit.edu

**David Sontag**
MIT
dsontag@csail.mit.edu

## Abstract

Recent attempts to achieve fairness in predictive models focus on the balance between fairness and accuracy. In sensitive applications such as healthcare or criminal justice, this trade-off is often undesirable as any increase in prediction error could have devastating consequences. In this work, we argue that the fairness of predictions should be evaluated in context of the data, and that unfairness induced by inadequate samples sizes or unmeasured predictive variables should be addressed through data collection, rather than by constraining the model. We decompose cost-based metrics of discrimination into bias, variance, and noise, and propose actions aimed at estimating and reducing each term. Finally, we perform case-studies on prediction of income, mortality, and review ratings, confirming the value of this analysis. We find that data collection is often a means to reduce discrimination without sacrificing accuracy.

## 1   Introduction

As machine learning algorithms increasingly affect decision making in society, many have raised concerns about the fairness and biases of these algorithms, especially in applications to healthcare or criminal justice, where human lives are at stake (Angwin et al., 2016; Barocas & Selbst, 2016). It is often hoped that the use of automatic decision support systems trained on observational data will remove human bias and improve accuracy. However, factors such as data quality and model choice may encode unintentional discrimination, resulting in systematic disparate impact.

We study fairness in prediction of outcomes such as recidivism, annual income, or patient mortality. Fairness is evaluated with respect to *protected groups* of individuals defined by attributes such as gender or ethnicity (Ruggieri et al., 2010). Following previous work, we measure discrimination in terms of differences in prediction cost across protected groups (Calders & Verwer, 2010; Dwork et al., 2012; Feldman et al., 2015). Correcting for issues of data provenance and historical bias in labels is outside of the scope of this work. Much research has been devoted to constraining models to satisfy cost-based fairness in prediction, as we expand on below. *The impact of data collection on discrimination has received comparatively little attention.*

Fairness in prediction has been encouraged by adjusting models through regularization (Bechavod & Ligett, 2017; Kamishima et al., 2011), constraints (Kamiran et al., 2010; Zafar et al., 2017), and representation learning (Zemel et al., 2013). These attempts can be broadly categorized as model-based approaches to fairness. Others have applied data preprocessing to reduce discrimination (Hajian & Domingo-Ferrer, 2013; Feldman et al., 2015; Calmon et al., 2017). For an empirical comparison, see for example Friedler et al. (2018). Inevitably, however, restricting the model class or perturbing training data to improve fairness may harm predictive accuracy (Corbett-Davies et al., 2017).

A *tradeoff* of predictive accuracy for fairness is sometimes difficult to motivate when predictions influence high-stakes decisions. In particular, post-hoc correction methods based on randomizing predictions (Hardt et al., 2016; Pleiss et al., 2017) are unjustifiable for ethical reasons in clinical tasks

such as severity scoring. Moreover, as pointed out by Woodworth et al. (2017), post-hoc correction may lead to suboptimal predictive accuracy compared to other equally fair classifiers.

Disparate predictive accuracy can often be explained by insufficient or skewed sample sizes or inherent unpredictability of the outcome given the available set of variables. With this in mind, we propose that fairness of predictive models should be analyzed in terms of model bias, model variance, and outcome noise *before* they are constrained to satisfy fairness criteria. This exposes and separates the adverse impact of inadequate data collection and the choice of the model on fairness. The cost of fairness need not always be one of predictive accuracy, but one of investment in data collection and model development. In high-stakes applications, the benefits often outweigh the costs.

In this work, we use the term "discrimination" to refer to specific kinds of differences in the predictive power of models when applied to different protected groups. In some domains, such differences may not be considered discriminatory, and it is critical that decisions made based on this information are sensitive to this fact. For example, in prior work, researchers showed that causal inference may help uncover which sources of differences in predictive accuracy introduce unfairness (Kusner et al., 2017). In this work, we assume that observed differences are considered discriminatory and discuss various means of explaining and reducing them.

**Main contributions**   We give a procedure for analyzing discrimination in predictive models with respect to cost-based definitions of group fairness, emphasizing the impact of data collection. First, we propose the use of bias-variance-noise decompositions for separating sources of discrimination. Second, we suggest procedures for estimating the value of collecting additional training samples. Finally, we propose the use of clustering for identifying subpopulations that are discriminated against to guide additional variable collection. We use these tools to analyze the fairness of common learning algorithms in three tasks: predicting income based on census data, predicting mortality of patients in critical care, and predicting book review ratings from text. We find that the accuracy in predictions of the mortality of cancer patients vary by as much as $20\%$ between protected groups. In addition, our experiments confirm that discrimination level is sensitive to the quality of the training data.

## 2   Background

We study fairness in prediction of an outcome $Y \in \mathcal{Y}$. Predictions are based on a set of covariates $X \in \mathcal{X} \subseteq \mathbb{R}^k$ and a *protected attribute $A \in \mathcal{A}$*. In mortality prediction, $X$ represents the medical history of a patient in critical care, $A$ the self-reported ethnicity, and $Y$ mortality. A model is considered fair if its errors are distributed similarly across protected groups, as measured by a cost function $\gamma$. Predictions learned from a training set $d$ are denoted $\hat{Y}_d := h(X, A)$ for some $h : \mathcal{X} \times \mathcal{A} \to \mathcal{Y}$ from a class $\mathcal{H}$. The protected attribute is assumed to be binary, $\mathcal{A} = \{0, 1\}$, but our results generalize to the non-binary case. A dataset $d = \{(x_i, a_i, y_i)\}_{i=1}^n$ consists of $n$ samples distributed according to $p(X, A, Y)$. When clear from context, we drop the subscript from $\hat{Y}_d$.

A popular cost-based definition of fairness is the *equalized odds* criterion, which states that a binary classifier $\hat{Y}$ is fair if its false negative rates (FNR) and false positive rates (FPR) are equal across groups (Hardt et al., 2016). We define FPR and FNR with respect to protected group $a \in \mathcal{A}$ by

$$\mathrm{FPR}_a(\hat{Y}) := \mathbb{E}_X[\hat{Y} \mid Y = 0, A = a], \quad \mathrm{FNR}_a(\hat{Y}) := \mathbb{E}_X[1 - \hat{Y} \mid Y = 1, A = a] \,.$$

Exact equality, $\mathrm{FPR}_0(\hat{Y}) = \mathrm{FPR}_1(\hat{Y})$, is often hard to verify or enforce in practice. Instead, we study the *degree* to which such constraints are violated. More generally, we use differences in *cost functions $\gamma_a$* between protected groups $a \in \mathcal{A}$ to define the *level of discrimination* $\Gamma$,

$$\Gamma^\gamma(\hat{Y}) := \left| \gamma_0(\hat{Y}) - \gamma_1(\hat{Y}) \right| \,. \tag{1}$$

In this work we study cost functions $\gamma_a \in \{\mathrm{FPR}_a, \mathrm{FNR}_a, \mathrm{ZO}_a\}$ in binary classification tasks, with $\mathrm{ZO}_a(\hat{Y}) := \mathbb{E}_X[\mathbb{1}[\hat{Y} \neq Y] \mid A = a]$ the *zero-one loss*. In regression problems, we use the group-specific *mean-squared error* $\mathrm{MSE}_a := \mathbb{E}_X[(\hat{Y} - Y)^2 \mid A = a]$. According to (1), predictions $\hat{Y}$ satisfy equalized odds on $d$ if $\Gamma^{\mathrm{FPR}}(\hat{Y}) = 0$ *and* $\Gamma^{\mathrm{FNR}}(\hat{Y}) = 0$.

**Calibration and impossibility**   A score-based classifier is *calibrated* if the prediction score assigned to a unit equals the fraction of positive outcomes for all units assigned similar scores. It

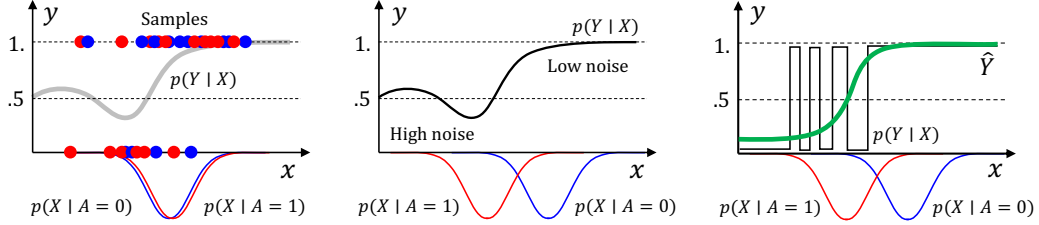

(a) For identically distributed protected groups and unaware outcome (see below), bias and noise are equal in expectation. Perceived discrimination is only due to variance.

(b) Heteroskedastic noise, i.e. $\exists x, x' : N(x) \neq N(x')$, may contribute to discrimination even for an optimal model if protected groups are not identically distributed.

(c) One choice of model may be more suited for one protected group, even under negligible noise and variance, resulting in a difference in expected bias, $\overline{B}_0 \neq \overline{B}_1$.

Figure 1: Scenarios illustrating how properties of the training set and model choice affect perceived discrimination in a binary classification task, under the assumption that outcomes and predictions are *unaware*, i.e. $p(Y \mid X, A) = p(Y \mid X)$ and $p(\hat{Y} \mid X, A) = p(\hat{Y} \mid X)$. Through bias-variance-noise decompositions (see Section 3.1), we can identify which of these dominate in their effect on fairness. We propose procedures for addressing each component in Section 4, and use them in experiments (see Section 5) to mitigate discrimination in income prediction and prediction of ICU mortality.

is impossible for a classifier to be calibrated in every protected group and satisfy multiple cost-based fairness criteria at once, unless accuracy is perfect or base rates of outcomes are equal across groups (Chouldechova, 2017). A relaxed version of this result (Kleinberg et al., 2016) applies to the discrimination level $\Gamma$. Inevitably, both constraint-based methods and our approach are faced with a choice between which fairness criteria to satisfy, and at what cost.

## 3 Sources of perceived discrimination

There are many potential sources of discrimination in predictive models. In particular, the choice of hypothesis class $\mathcal{H}$ and learning objective has received a lot of attention (Calders & Verwer, 2010; Zemel et al., 2013; Fish et al., 2016). However, data collection—the chosen set of predictive variables $X$, the sampling distribution $p(X, A, Y)$, and the training set size $n$—is an equally integral part of deploying fair machine learning systems in practice, and it should be guided to promote fairness. Below, we tease apart sources of discrimination through bias-variance-noise decompositions of cost-based fairness criteria. In general, we may think of noise in the outcome as the effect of a set of unobserved variables $U$, potentially interacting with $X$. Even the optimal achievable error for predictions based on $X$ may be reduced further by observing parts of $U$. In Figure 1, we illustrate three common learning scenarios and study their fairness properties through bias, variance, and noise.

To account for randomness in the sampling of training sets, we redefine discrimination level (1) in terms of the *expected* cost $\overline{\gamma}_a(\hat{Y}) := \mathbb{E}_D[\gamma_a(\hat{Y}_D)]$ over draws of a random training set $D$.

**Definition 1.** The *expected discrimination level* $\overline{\Gamma}(\hat{Y})$ of a predictive model $\hat{Y}$ learned from a random training set $D$, is

$$\overline{\Gamma}(\hat{Y}) := \left| \mathbb{E}_D \left[ \gamma_0(\hat{Y}_D) - \gamma_1(\hat{Y}_D) \right] \right| = \left| \overline{\gamma}_0(\hat{Y}) - \overline{\gamma}_1(\hat{Y}) \right| .$$

$\overline{\Gamma}(\hat{Y})$ is not observed in practice when only a single training set $d$ is available. If $n$ is small, it is recommended to estimate $\overline{\Gamma}$ through re-sampling methods such as bootstrapping (Efron, 1992).

### 3.1 Bias-variance-noise decompositions of discrimination level

An algorithm that learns models $\hat{Y}_D$ from datasets $D$ is given, and the covariates $X$ and size of the training data $n$ are fixed. We assume that $\hat{Y}_D$ is a deterministic function $\hat{y}_D(x, a)$ given the training set $D$, e.g. a thresholded scoring function. Following Domingos (2000), we base our analysis on decompositions of loss functions $L$ evaluated at points $(x, a)$. For decompositions of costs $\gamma_a \in \{\text{ZO}, \text{FPR}, \text{FNR}\}$ we let this be the zero-one loss, $L(y, y') = \mathbb{1}[y \neq y']$, and for

$\gamma_a$ = MSE, the squared loss, $L(y, y') = (y - y')^2$. We define the *main prediction* $\tilde{y}(x, a) = \arg\min_{y'} \mathbb{E}_D[L(\hat{Y}_D, y') \mid X = x, A = a]$ as the average prediction over draws of training sets for the squared loss, and the majority vote for the zero-one loss. The *(Bayes) optimal prediction* $y^*(x, a) = \arg\min_{y'} \mathbb{E}_Y[L(Y, y') \mid X = x, A = a]$ achieves the smallest expected error with respect to the random outcome $Y$.

**Definition 2** (Bias, variance and noise). Following Domingos (2000), we define bias $B$, variance $V$ and noise $N$ at a point $(x, a)$ below.

$$B(\hat{Y}, x, a) = L(y^*(x, a), \tilde{y}(x, a)) \qquad N(x, a) = \mathbb{E}_Y[L(y^*(x, a), Y) \mid X = x, A = a]$$
$$V(\hat{Y}, x, a) = \mathbb{E}_D[L(\tilde{y}(x, a), \hat{y}_D(x, a))] . \tag{2}$$

Here, $y^*, \hat{y}$ and $\tilde{y}$, are all deterministic functions of $(x, a)$, while $Y$ is a random variable.

In words, the bias $B$ is the loss incurred by the main prediction relative to the optimal prediction. The variance $V$ is the average loss incurred by the predictions learned from different datasets relative to the main prediction. The noise $N$ is the remaining loss independent of the learning algorithm, often known as the Bayes error. We use these definitions to decompose $\overline{\Gamma}$ under various definitions of $\gamma_a$.

**Theorem 1.** *With $\overline{\gamma}_a$ the group-specific zero-one loss or class-conditional versions (e.g. FNR, FPR), or the mean squared error, $\overline{\gamma}_a$ and the discrimination level $\overline{\Gamma}$ admit decompositions of the form*

$$\overline{\gamma}_a(\hat{Y}) = \underbrace{\overline{N}_a}_{Noise} + \underbrace{\overline{B}_a(\hat{Y})}_{Bias} + \underbrace{\overline{V}_a(\hat{Y})}_{Variance} \quad and \quad \overline{\Gamma} = \left| (\overline{N}_0 - \overline{N}_1) + (\overline{B}_0 - \overline{B}_1) + (\overline{V}_0 - \overline{V}_1) \right|$$

*where we leave out $\hat{Y}$ in the decomposition of $\overline{\Gamma}$ for brevity. With $B, V$ defined as in (2), we have*

$$\overline{B}_a(\hat{Y}) = \mathbb{E}_X[B(\tilde{y}, X, a) \mid A = a] \quad and \quad \overline{V}_a(\hat{Y}) = \mathbb{E}_{X,D}[c_v(X)V(\hat{Y}_D, X, a) \mid A = a] .$$

*For the zero-one loss, $c_v(x, a) = 1$ if $\hat{y}_m(x, a) = y^*(x, a)$, otherwise $c_v(x, a) = -1$. For the squared loss $c_v(x, a) = 1$. The noise term for population losses is*

$$\overline{N}_a := \mathbb{E}_X[c_n(X, a)L(y^*(X, a), Y) \mid A = a]$$

*and for class-conditional losses w.r.t class $y \in \{0, 1\}$,*

$$\overline{N}_a(y) := \mathbb{E}_X[c_n(X, a)L(y^*(X, a), y) \mid A = a, Y = y] .$$

*For the zero-one loss, and class-conditional variants, $c_n(x, a) = 2\mathbb{E}_D[\mathbb{1}[\hat{y}_D(x, a) = y^*(x, a)]] - 1$ and for the squared loss, $c_n(x, a) = 1$.*

*Proof sketch.* Conditioning and exchanging order of expectation, the cases of mean squared error and zero-one losses follow from Domingos (2000). Class-conditional losses follow from a case-by-case analysis of possible errors. See the supplementary material for a full proof. □

Theorem 1 points to distinct sources of perceived discrimination. Significant differences in bias $\overline{B}_0 - \overline{B}_1$ indicate that the chosen model class is not flexible enough to fit both protected groups well (see Figure 1c). This is typical of (misspecified) linear models which approximate non-linear functions well only in small regions of the input space. Regularization or post-hoc correction of models effectively increase the bias of one of the groups, and should be considered only if there is reason to believe that the original bias is already minimal.

Differences in variance, $\overline{V}_0 - \overline{V}_1$, could be caused by differences in sample sizes $n_0, n_1$ or group-conditional feature variance $\text{Var}(X \mid A)$, combined with a high capacity model. Targeted collection of training samples may help resolve this issue. Our decomposition does not apply to post-hoc randomization methods (Hardt et al., 2016) but we may treat these in the same way as we do random training sets and interpret them as increasing the variance $\overline{V}_a$ of one group to improve fairness.

When noise is significantly different between protected groups, discrimination is partially unrelated to model choice and training set size and may only be reduced by measuring additional variables.

**Proposition 1.** *If $\overline{N}_0 \neq \overline{N}_1$, no model can be 0-discriminatory in expectation without access to additional information or increasing bias or variance w.r.t. to the Bayes optimal classifier.*

*Proof.* By definition, $\overline{\Gamma} = 0 \implies (\overline{N}_1 - \overline{N}_0) = (\overline{B}_0 - \overline{B}_1) + (\overline{V}_0 - \overline{V}_1)$. As the Bayes optimal classifier has neither bias nor variance, the result follows immediately. $\qquad\square$

In line with Proposition 1, most methods for ensuring algorithmic fairness reduce discrimination by trading off a difference in noise for one in bias or variance. However, this trade-off is only motivated if the considered predictive model is close to Bayes optimal *and* no additional predictive variables may be measured. Moreover, if noise is homoskedastic in regression settings, post-hoc randomization is ill-advised, as the difference in Bayes error $\overline{N}_0 - \overline{N}_1$ is zero, and discrimination is caused only by model bias or variance (see the supplementary material for a proof).

**Estimating bias, variance and noise**   Group-specific variance $\overline{V}_a$ may be estimated through sample splitting or bootstrapping (Efron, 1992). In contrast, the noise $\overline{N}_a$ and bias $\overline{B}_a$ are difficult to estimate when $X$ is high-dimensional or continuous. In fact, no convergence results of noise estimates may be obtained without further assumptions on the data distribution (Antos et al., 1999). Under some such assumptions, noise may be approximately estimated using distance-based methods (Devijver & Kittler, 1982), nearest-neighbor methods (Fukunaga & Hummels, 1987; Cover & Hart, 1967), or classifier ensembles (Tumer & Ghosh, 1996). When comparing the discrimination level of two different models, noise terms cancel, as they are independent of the model. As a result, *differences* in bias may be estimated even when the noise is not known (see the supplementary material).

**Testing for significant discrimination**   When sample sizes are small, perceived discrimination may not be statistically significant. In the supplementary material, we give statistical tests both for the discrimination level $\Gamma(\hat{Y})$ and the difference in discrimination level between two models $\hat{Y}, \hat{Y}'$.

## 4   Reducing discrimination through data collection

In light of the decomposition of Theorem 1, we explore avenues for reducing group differences in bias, variance, and noise without sacrificing predictive accuracy. In practice, predictive accuracy is often artificially limited when data is expensive or impractical to collect. With an investment in training samples or measurement of predictive variables, both accuracy and fairness may be improved.

### 4.1   Increasing training set size

Standard regularization used to avoid overfitting is not guaranteed to improve or preserve fairness. An alternative route is to collect more training samples and reduce the impact of the bias-variance trade-off. When supplementary data is collected from the same distribution as the existing set, covariate shift may be avoided (Quionero-Candela et al., 2009). This is often achievable; labeled data may be expensive, such as when paying experts to label observations, but given the means to acquire additional labels, they would be drawn from the original distribution. To estimate the value of increasing sample size, we predict the discrimination level $\overline{\Gamma}(\hat{Y}_D)$ as $D$ increases in size.

The curve measuring generalization performance of predictive models as a function of training set size $n$ is called a Type II *learning curve* (Domhan et al., 2015). We call $\overline{\gamma}_a(\hat{Y}, n) := \mathbb{E}[\gamma_a(\hat{Y}_{D_n})]$, as a function of $n$, the learning curve with respect to protected group $a$. We define the discrimination learning curve $\overline{\Gamma}(\hat{Y}, n) := |\overline{\gamma}_0(\hat{Y}, n) - \overline{\gamma}_1(\hat{Y}, n)|$ (see Figure 2a for an example). Empirically, learning curves behave asymptotically as *inverse power-law* curves for diverse algorithms such as deep neural networks, support vector machines, and nearest-neighbor classifiers, even when model capacity is allowed to grow with $n$ (Hestness et al., 2017; Mukherjee et al., 2003). This observation is also supported by theoretical results (Amari, 1993).

**Assumption 1** (Learning curves)**.** *The population prediction loss $\overline{\gamma}(\hat{Y}, n)$, and group-specific losses $\overline{\gamma}_0(\hat{Y}, n), \overline{\gamma}_1(\hat{Y}, n)$, for a fixed learning algorithm $\hat{Y}$, behave asymptotically as inverse power-law curves with parameters $(\alpha, \beta, \delta)$. That is, $\exists M, M_0, M_1$ such that for $n \geq M, n_a \geq M_a$,*

$$\overline{\gamma}(\hat{Y}, n) = \alpha n^{-\beta} + \delta \quad and \quad \forall a \in \mathcal{A} : \overline{\gamma}_a(\hat{Y}, n_a) = \alpha_a n_a^{-\beta_a} + \delta_a \qquad (3)$$

Intercepts, $\delta, \delta_a$ in (3) represent the asymptotic bias $\overline{B}(\hat{Y}_{D_\infty})$ and the Bayes error $\overline{N}$, with the former vanishing for consistent estimators. Accurately estimating $\delta$ from finite samples is often challenging as the first term tends to dominate the learning curve for practical sample sizes.

In experiments, we find that the inverse power-laws model fit group conditional ($\gamma_a$) and class-conditional (FPR, FNR) errors well, and use these to extrapolate $\overline{\Gamma}(\hat{Y}, n)$ based on estimates from subsampled data.

## 4.2 Measuring additional variables

When discrimination $\overline{\Gamma}$ is dominated by a difference in noise, $\overline{N}_0 - \overline{N}_1$, fairness may not be improved through model selection alone without sacrificing accuracy (see Proposition 1). Such a scenario is likely when available covariates are not equally predictive of the outcome in both groups. We propose identification of clusters of individuals in which discrimination is high as a means to guide further variable collection—if the variance in outcomes within a cluster is not explained by the available feature set, additional variables may be used to further distinguish its members.

Let a random variable $C$ represent a (possibly stochastic) clustering such that $C = c$ indicates membership in cluster $c$. Then let $\rho_a(c)$ denote the expected prediction cost for units in cluster $c$ with protected attribute $a$. As an example, for the zero-one loss we let

$$\rho_a^{\text{ZO}}(c) := \mathbb{E}_X[\mathbb{1}[\hat{Y} \neq Y] \mid A = a, C = c],$$

and define $\rho$ analogously for false positives or false negatives. Clusters $c$ for which $|\rho_0(c) - \rho_1(c)|$ is large identify groups of individuals for which discrimination is worse than average, and can guide targeted collection of additional variables or samples. In our experiments on income prediction, we consider particularly simple clusterings of data defined by subjects with measurements above or below the average value of a single feature $x(c)$ with $c \in \{1, \ldots, k\}$. In mortality prediction, we cluster patients using topic modeling. As measuring additional variables is expensive, the utility of a candidate set should be estimated before collecting a large sample (Koepke & Bilenko, 2012).

## 5 Experiments

We analyze the fairness properties of standard machine learning algorithms in three tasks: prediction of income based on national census data, prediction of patient mortality based on clinical notes, and prediction of book review ratings based on review text.[1] We disentangle sources of discrimination by assessing the level of discrimination for the full data, estimating the value of increasing training set size by fitting Type II learning curves, and using clustering to identify subgroups where discrimination is high. In addition, we estimate the Bayes error through non-parametric techniques.

In our experiments, we omit the sensitive attribute $A$ from our classifiers to allow for closer comparison to previous works, e.g. Hardt et al. (2016); Zafar et al. (2017). In preliminary results, we found that fitting separate classifiers for each group increased the error rates of both groups due to the resulting smaller sample size, as classifiers could not learn from other groups. As our model objective is to maximize accuracy over all data points, our analysis uses a single classifier trained on the entire population.

### 5.1 Income prediction

Predictions of a person's salary may be used to help determine an individual's market worth, but systematic underestimation of the salary of protected groups could harm their competitiveness on the job market. The Adult dataset in the UCI Machine Learning Repository (Lichman, 2013) contains 32,561 observations of yearly income (represented as a binary outcome: over or under $50,000) and twelve categorical or continuous features including education, age, and marital status. Categorical attributes are dichotomized, resulting in a total of 105 features.

We follow Pleiss et al. (2017) and strive to ensure fairness across genders, which is excluded as a feature from the predictive models. Using an 80/20 train-test split, we learn a random forest predictor, which is is well-calibrated for both groups (Brier (1950) scores of 0.13 and 0.06 for men and women). We find the difference in zero-one loss $\Gamma^{\text{ZO}}(\hat{Y})$ has a 95%-confidence interval[2] $.085 \pm .069$ with decision thresholds at 0.5. At this threshold, the false negative rates are $0.388 \pm 0.026$ and $0.448 \pm 0.064$ for men and women respectively, and the false positive rates $0.111 \pm 0.011$ and

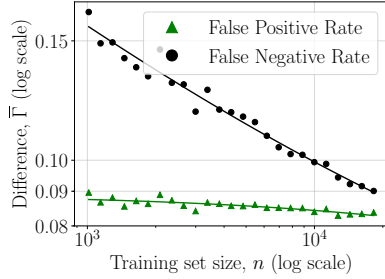

| Method | $E_{low}$ | $E_{up}$ | group |
|---|---|---|---|
| Mahalanobis | – | 0.29 | men |
| (Mahalanobis, 1936) | – | 0.13 | women |
| Bhattacharyya | 0.001 | 0.040 | men |
| (Bhattacharyya, 1943) | 0.001 | 0.027 | women |
| Nearest Neighbors | 0.10 | 0.19 | men |
| (Cover & Hart, 1967) | 0.04 | 0.07 | women |

(a) Group differences in false positive rates and false negative rates for a random forest classifier decrease with increasing training set size.

(b) Estimation of Bayes error lower and upper bounds ($E_{low}$ and $E_{up}$) for zero-one loss of men and women. Intervals for men and women are non-overlapping for Nearest Neighbors.

Figure 2: Discrimination level and noise estimation in income prediction with the Adult dataset.

$0.033 \pm 0.008$. We focus on random forest classifiers, although we found similar results for logistic regression and decision trees.

We examine the effect of varying training set size $n$ on discrimination. We fit inverse power-law curves to estimates of $\text{FPR}(\hat{Y}, n)$ and $\text{FNR}(\hat{Y}, n)$ using repeated sample splitting where at least 20% of the full data is held out for evaluating generalization error at every value of $n$. We tune hyperparameters for each training set size for decision tree classifiers and logistic regression but tuned over the entire dataset for random forest. We include full training details in the supplementary material. Metrics are averaged over 50 trials. See Figure 2a for the results for random forests. Both FPR and FNR decrease with additional training samples. The discrimination level $\Gamma^{\text{FNR}}$ for false negatives decreases by a striking 40% when increasing the training set size from 1000 to 10,000. This suggests that trading off accuracy for fairness at small sample sizes may be ill-advised. Based on fitted power-law curves, we estimate that for unlimited training data drawn from the same distribution, we would have $\Gamma^{\text{FNR}}(\hat{Y}) \approx 0.04$ and $\Gamma^{\text{FPR}}(\hat{Y}) \approx 0.08$.

In Figure 2b, we compare estimated upper and lower bounds on noise ($E_{low}$ and $E_{up}$) for men and women using the Mahalanobis and Bhattacharyya distances (Devijver & Kittler, 1982), and a $k$-nearest neighbor method (Cover & Hart, 1967) with $k = 5$ and 5-fold cross validation. Men have consistently higher noise estimates than women, which is consistent with the differences in zero-one loss found using all models. For nearest neighbors estimates, intervals for men and women are non-overlapping, which suggests that noise may contribute substantially to discrimination.

To guide attempts at reducing discrimination further, we identify clusters of individuals for whom false negative predictions are made at different rates between protected groups, with the method described in Section 4.2. We find that for individuals in executive or managerial occupations (12% of the sample), false negatives are more than twice as frequent for women (0.412) as for men (0.157). For individuals in all other occupations, the difference is significantly smaller, 0.543 for women and 0.461 for men, despite the fact that the disparity in outcome base rates in this cluster is large (0.26 for men versus 0.09 for women). A possible reason is that in managerial occupations the available variable set explains a larger portion of the variance in salary for men than for women. If so, further sub-categorization of managerial occupations could help reduce discrimination in prediction.

### 5.2 Intensive care unit mortality prediction

Unstructured medical data such as clinical notes can reveal insights for questions like mortality prediction; however, disparities in predictive accuracy may result in discrimination of protected groups. Using the MIMIC-III dataset of all clinical notes from 25,879 adult patients from Beth Israel Deaconess Medical Center (Johnson et al., 2016), we predict hospital mortality of patients in critical care. Fairness is studied with respect to five self-reported ethnic groups of the following proportions: Asian (2.2%), Black (8.8%), Hispanic (3.4%), White (70.8%), and Other (14.8%). Notes were collected in the first 48 hours of an intensive care unit (ICU) stay; discharge notes were excluded. We only included patients that stayed in the ICU for more than 48 hours. We use the tf-idf statistics of the 10,000 most frequent words as features. Training a model on 50% of the data, selecting

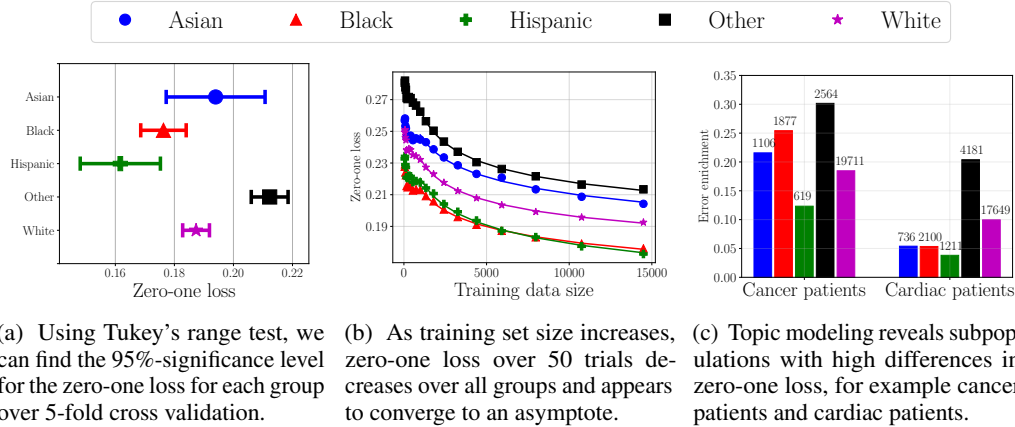

(a) Using Tukey's range test, we can find the 95%-significance level for the zero-one loss for each group over 5-fold cross validation.

(b) As training set size increases, zero-one loss over 50 trials decreases over all groups and appears to converge to an asymptote.

(c) Topic modeling reveals subpopulations with high differences in zero-one loss, for example cancer patients and cardiac patients.

Figure 3: Mortality prediction from clinical notes using logistic regression. Best viewed in color.

hyper-parameters on 25%, and testing on 25%, we find that logistic regression with L1-regularization achieves an AUC of 0.81. The logistic regression is well-calibrated with Brier scores ranging from 0.06-0.11 across the five groups; we note better calibration is correlated with lower prediction error.

We report cost and discrimination level in terms of generalized zero-one loss (Pleiss et al., 2017). Using an ANOVA test (Fisher, 1925) with $p < 0.001$, we reject the null hypothesis that loss is the same among all five groups. To map the 95% confidence intervals, we perform pairwise comparisons of means using Tukey's range test (Tukey, 1949) across 5-fold cross-validation. As seen in Figure 3a, patients in the Other and Hispanic groups have the highest and lowest generalized zero-one loss, respectively, with relatively few overlapping intervals. Notably, the largest ethnic group (White) does not have the best accuracy, whereas smaller ethnic groups tend towards extremes. While racial groups differ in hospital mortality base rates (Table 1 in the Supplementary material), Hispanic (10.3%) and Black (10.9%) patients have very different error rates despite similar base rates.

To better understand the discrimination induced by our model, we explore the effect of changing training set size. To this end, we repeatedly subsample and split the data, holding out at least 20% of the full data for testing. In Figure 3b, we show loss averaged over 50 trials of training a logistic regression on increasingly larger training sets; estimated inverse power-law curves show good fits. We see that some pairwise differences in loss decrease with additional training data.

Next, we identify clusters for which the difference in prediction errors between protected groups is large. We learn a topic model with $k = 50$ topics generated using Latent Dirichlet Allocation (Blei et al., 2003). Topics are concatenated into an $n \times k$ matrix $Q$ where $q_{ic}$ designates the proportion of topic $c \in [k]$ in note $i \in [n]$. Following prior work on enrichment of topics in clinical notes (Marlin et al., 2012; Ghassemi et al., 2014), we estimate the probability of patient mortality $Y$ given a topic $c$ as $\hat{p}(Y|C = c) := (\sum_{i=1}^{n} y_i q_{ic})/(\sum_{i=1}^{n} q_{ic})$ where $y_i$ is the hospital mortality of patient $i$. We compare relative error rates given protected group and topic using binary predicted mortality $\hat{y}_i$, actual mortality $y_i$, and group $a_i$ for patient $i$ through

$$\hat{p}(\hat{Y} \neq Y \mid A = a', C = c) = \frac{\sum_{i=1}^{n} \mathbb{1}(y_i \neq \hat{y}_i)\mathbb{1}(a_i = a')q_{ic}}{\sum_{i=1}^{n} \mathbb{1}(a_i = a')q_{ic}}$$

which follows using substitution and conditioning on $A$. These error rates were computed using a logistic regression with L1 regularization using an 80/20 train-test split over 50 trials. While many topics have consistent error rates across groups, some topics (e.g. cardiac patients or cancer patients as shown in Figure 3c) have large differences in error rates across groups. We include more detailed topic descriptions in the supplementary material. Once we have identified a subpopulation with particularly high error, for example cancer patients, we can consider collecting more features or collecting more data from the same data distribution. We find that error rates differ between 0.12 and 0.30 across protected groups of cancer patients, and between 0.05 and 0.20 for cardiac patients.

### 5.3 Book review ratings

In the supplementary material, we study prediction of book review ratings from review texts (Gnanesh, 2017). The protected attribute was chosen to be the gender of the author as determined from Wikipedia. In the dataset, the difference in mean-squared error $\Gamma^{\mathrm{MSE}}(\hat{Y})$ has 95%-confidence interval $0.136 \pm 0.048$ with $\mathrm{MSE}_M = 0.224$ for reviews for male authors and $\mathrm{MSE}_F = 0.358$. Strikingly, our findings suggest that $\Gamma^{\mathrm{MSE}}(\hat{Y})$ may be completely eliminated by additional targeted sampling of the less represented gender.

## 6 Discussion

We identify that existing approaches for reducing discrimination induced by prediction errors may be unethical or impractical to apply in settings where predictive accuracy is critical, such as in healthcare or criminal justice. As an alternative, we propose a procedure for analyzing the different sources contributing to discrimination. Decomposing well-known definitions of cost-based fairness criteria in terms of differences in bias, variance, and noise, we suggest methods for reducing each term through model choice or additional training data collection. Case studies on three real-world datasets confirm that collection of additional samples is often sufficient to improve fairness, and that existing post-hoc methods for reducing discrimination may unnecessarily sacrifice predictive accuracy when other solutions are available.

Looking forward, we can see several avenues for future research. In this work, we argue that identifying clusters or subpopulations with high predictive disparity would allow for more targeted ways to reduce discrimination. We encourage future research to dig deeper into the question of local or context-specific unfairness in general, and into algorithms for addressing it. Additionally, extending our analysis to intersectional fairness (Buolamwini & Gebru, 2018; Hébert-Johnson et al., 2017), e.g. looking at both gender and race or all subdivisions, would provide more nuanced grappling with unfairness. Finally, additional data collection to improve the model may cause unexpected delayed impacts (Liu et al., 2018) and negative feedback loops (Ensign et al., 2017) as a result of distributional shifts in the data. More broadly, we believe that the study of fairness in non-stationary populations is an interesting direction to pursue.

## Acknowledgements

The authors would like to thank Yoni Halpern and Hunter Lang for helpful comments, and Zeshan Hussain for clinical guidance. This work was partially supported by Office of Naval Research Award No. N00014-17-1-2791 and NSF CAREER award #1350965.

## Footnotes

[1] A synthetic experiment validating group-specific learning curves is left to the supplementary material.

[2] Details for computing statistically significant discrimination can be found in the supplementary material.

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
