[Supplementary Material]

# Supplementary Material for
# Why Is My Classifier Discriminatory?

**Irene Y. Chen**
MIT
iychen@mit.edu

**Fredrik D. Johansson**
MIT
fredrikj@mit.edu

**David Sontag**
MIT
dsontag@csail.mit.edu

## 1  Testing for significant discrimination

In general, neither $\Gamma$ nor $\overline{\Gamma}$ can be computed exactly, as the expectations $\gamma_a = \mathbb{E}_p[L(Y, \hat{Y}) \mid A = a]$ and $\overline{\gamma}$, for $a \in \mathcal{A}$ are known only approximately through a set of samples $S = \{(x_i, a_i, y_i)\}_{i=1}^m \sim p^m$ drawn from the (possibly class-conditional) population $p$. The Monte Carlo estimate,

$$\gamma_a^S(\hat{Y}) = \frac{1}{m_a} \sum_{i=1}^m L(y_i, \hat{y}_i) \mathbb{1}[a_i = a] \,,$$

with $m_a = \sum_{i=1}^m \mathbb{1}[a_i = a]$, may be used to form an estimate $\Gamma^S(\hat{Y}) = |\gamma_0^S(\hat{Y}) - \gamma_1^S(\hat{Y})|$. By the central limit theorem, for sufficiently large $m$, $\gamma_a^S(\hat{Y}) \sim \mathcal{N}(\mu_a, \sigma_a^2/m_a)$ and $(\gamma_0^S - \gamma_1^S) \sim \mathcal{N}(\mu_0 - \mu_1, \sigma_0^2/m_0 + \sigma_1^2/m_1)$. As a result, the significance of $\Gamma^S(\hat{Y})$ can be tested with a two-tailed z-test or using the test of Woodworth et al. (2017). If sample sizes are small and the target binary, more appropriate tests are available (Brown et al., 2001). In addition, we will often want to compare the discrimination levels $\Gamma(\hat{Y}), \Gamma(\hat{Y}')$ of predictors $\hat{Y}, \hat{Y}'$, resulting from different learning algorithms, models, or sets of observed variables. The random variable $|\Gamma^S(\hat{Y}) - \Gamma^S(\hat{Y}')|$ is not Normal distributed, but is an absolute difference of folded-normal variables. However, for any $\alpha \in \{-1, 1\}$, $Z_\alpha := \alpha(\gamma_0^S(\hat{Y}) - \gamma_1^S(\hat{Y})) - (\gamma_0^S(\hat{Y}') - \gamma_1^S(\hat{Y}'))$ is Normal distributed. Further, by enumerating the signs of $(\gamma_0^S(\hat{Y}) - \gamma_1^S(\hat{Y}))$ and $(\gamma_0^S(\hat{Y}') - \gamma_1^S(\hat{Y}'))$, we can show that $|\Gamma^S - \Gamma^{S'}| = \min_{\alpha \in \{-1,1\}} |Z_\alpha|$. As a result, to reject the null hypothesis $H_0 : \Gamma = \Gamma'$, we require that the observed values of both $Z_{-1}$ and $Z_1$ are unlikely under $H_0$ at given significance.

## 2  Additional experimental details

### 2.1  Datasets

- Adult Income Dataset (Lichman, 2013). The dataset has 32,561 instances. The target variable indicates whether or not income is larger than 50K dollars, and the sensitive feature is Gender. Each data object is described by 14 attributes which include 8 categorical and 6 numerical attributes. We quantize the categorical attributes into binary features and keep the continuous attributes, which results in 105 features for prediction. We note the label imbalance as 30% of male adults have income over 50K whereas only 10% of female adults have income over 50K. Additionally 24% of all adults have salary over 50K, and the dataset has 33% women and 67% men.

- Goodreads reviews Gnanesh (2017), only included in the supplemental materials. The dataset was collected from Oct 12, 2017 to Oct 21, 2017 and has 13,244 reviews. The target variable is the rating of the review, and the sensitive feature is the gender of the author. Genders were gathered by querying Wikipedia and using pronoun inference, and the dataset is a subset of the original Goodreads dataset because it only includes reviews about the

| (a) Learning curves, $\overline{\gamma}_0, \overline{\gamma}_1$ for random forest | (b) Discrimination, $\overline{\Gamma} = |\overline{\gamma}_0 - \overline{\gamma}_1|$ for various models |

Figure 1: Inverse power-laws (Pow3) fit to generalization error as a function of training set size on synthetic data. Dotted lines are extrapolations from sample sizes indicated by black stars. This illustrates the difficulty of estimating the Bayes error through extrapolation, here at $\overline{N}_0 = 3 \cdot 10^{-4}$ and $\overline{N}_1 = 7 \cdot 10^{-3}$ respectively.

    top 100 most popular authors. Each datum consists of the review text, vectorized using Tf-Idf. The review scores occurred with counts 578, 2606, 4544, 5516 for scores 1,3,4, and 5 respectively. Books by women authors and men authors had average scores of 4.088 and 4.092 respectively.

- MIMIC-III dataset (Johnson et al., 2016). The dataset includes 25,879 adult patients admitted to the intensive care unit of the Beth Israel Deaconess Medical Center in downtown Boston. Clinical notes from the first 48 hours are used to predict hospital mortality after 48 hours. Of all adult patients, 13.8% patients died in the hospital. We are interested in the difference in performance between the five self-reported ethnic groups and following data sizes and hospital mortality rates.

| Race | # patients | % total | Hospital Mortality |
|---|---|---|---|
| Asian | 583 | 2.3 | 14.2 |
| Black | 2,327 | 9.0 | 10.9 |
| Hispanic | 832 | 3.2 | 10.3 |
| Other | 3,761 | 14.5 | 18.4 |
| White | 18,377 | 71.0 | 13.4 |

Table 1: Summary statistics of clinical notes dataset

## 2.2 Synthetic experiments

To illustrate the effect of training set size and model choice, and the validity of the power-law learning curve assumption, we conduct a small synthetic experiment in which $p(A = 1) = 0.3$ and $X \sim \mathcal{N}(\mu_A, \sigma_A^2)$ with $\mu_0 = 0, \mu_1 = 1, \sigma_0 = 1, \sigma_1 = 2$. The outcome is a quadratic function with heteroskedastic noise, $Y = 2X^2 - 2X + .1 + \epsilon X^2$, with $\epsilon \sim \mathcal{N}(0, 1)$. We fit decision tree, random forest and ridge regressors of the outcome $Y$ to $X$ using default parameters in the implementation in scikit-learn (Pedregosa et al., 2011), but limiting the decision tree to depth $T \leq 4$. The size of the training set is varied exponentially between $2^5$ and $2^{17}$ samples, and at each size, trees are fit 200 times. In Figure 1, we show the resulting learning curves $\overline{\gamma}_0(\hat{Y}, n)$ and $\overline{\gamma}_1(\hat{Y}, n)$ as well as fits of Pow3 curves to them. Shown in dotted lines are extrapolations of learning curves from different sample sizes, illustrating the difficulty of estimating the intercepts $\delta_a$ and the Bayes error with high accuracy.

## 2.3 Book review ratings

Sentiment and rating prediction from text reveal quantitative insights from unstructured data; however deficiencies in algorithmic prediction may incorrectly represent populations. Using a dataset of 13,244 reviews collected from Goodreads (Gnanesh, 2017) with inferred author sex scraped from

(a) As training set size increases for random forest, MSE decreases but maintains difference between groups. Intercepts from fitted power-laws show no difference in noise.

(b) Holding number of reviews for male authors $n_M$ steady and varying number of reviews for female authors $n_F$, we can achieve higher MSE for one group than with the full dataset.

Figure 2: Goodreads dataset for book rating prediction. Adding training data decreases overall mean squared error (MSE) for both groups while adding training data to only one group has a much bigger impact on reducing $\overline{\Gamma}$. Increasing the number of features reduces MSE but does not reduce $\overline{\Gamma}$.

Wikipedia, we seek to predict the review rating based on the review text. We use as features the Tf-Idf statistics of the 5000 most frequent words. Our protected attribute is gender of the author of the book, and the target attribute is the rating (1-5) of the review. The data is heavily imbalanced, with 18% reviews about female authors versus 82% reviews about male authors.

We observe statistically significant levels of discrimination with respect to mean squared error (MSE) with linear regression, decision trees and random forests. Using a random forest and training on 80% of the dataset and testing on 20%, we find that our $\Gamma^{\mathrm{MSE}}(\hat{Y})$ has 95%-confidence interval $0.180 \pm 0.044$ with $\mathrm{MSE}_M = 0.314$ for reviews for male authors and $\mathrm{MSE}_F = 0.494$ for reviews for female authors using a difference in means statistical test. Results were found after hyperparameter turning for each training set size and taking an average over 50 trials. We observe similar patterns with linear regression and decision trees.

To estimate the impact of additional training data, we evaluate the effect of varying training set size $n$ on predictive performance and discrimination. Through repeated sample spitting, we train a random forest on increasing training set sizes, reserving at least 20% of the dataset for testing. In Figure 2a, additional training data lowers $\mathrm{MSE}_F$ and $\mathrm{MSE}_M$, fitting an inverse power-law. Based on the intercept terms of the extrapolated power-laws ($\delta_M = 0.0011$ for reviews with male authors and $\delta_F = 0.0013$ for reviews with female authors), we may expect that $\overline{\Gamma}$ can be explained more by differences in bias and variance than by noise since our estimated difference in noise $|\delta_F - \delta_M| \approx 0$.

In order to further measure the effect of collecting more samples, we analyze a one-sized increase in training data. Because of the initial skew of author genders in the dataset, we vary the number of reviews for female authors, creating a shift in populations in the training data. We fix the training set size of reviews for male authors at $n_M = 1939$, which represents the size of the full data for female authors $N_F$, reserving 20% of the dataset as test data. We then vary the training data size for female authors $n_F$ such that the ratio $n_F/n_M$ varies evenly between 0.1 to 1.0. Using a linear regression in Figure 2b, we see that as the ratio $n_F/n_M$ increases, $\mathrm{MSE}_F$ decreases far below $\mathrm{MSE}_M$ and far below our best reported MSE of the random forest on the full dataset. This suggests that shifting the data ratio and collecting more data for the under-represented group can adapt our model to reduce discrimination.

## 2.4 Clinical notes

Here we include additional details about topic modeling. Topics were sampled using Markov Chain Monte Carlo after 2,500 iterations. We present the topics with highest and lowest variance in error rates among groups in Table 2. Error rates were computed using a logistic regression with L1 regularization over 10,000 TF-IDF features using 80/20 training and testing data split over 50 trials.

Based on the most representative words for each topic, we can infer topic descriptions, for example cancer patients for topic 48 and cardiac patients for topic 45.

| Topic | Top words | Asian | Black | Hispanic | Other | White |
|---|---|---|---|---|---|---|
| 31 | no(t pain present normal edema tube history pulse absent left respiratory monitor | 5.9 | 8.4 | 17.6 | 30.8 | 11.1 |
| 17 | hospital lymphoma continue s/p unit bmt thrombocytopenia line rash | 34.3 | 13.6 | 34.9 | 30.2 | 26.0 |
| 43 | bowel abdominal abd abdomen surgery s/p small pain obstruction fluid ngt | 16.6 | 11.8 | 5.7 | 26.8 | 13.2 |
| 45 | artery carotid aneurysm left identifier numeric vertebral internal clip | 5.4 | 5.3 | 3.8 | 20.4 | 10.0 |
| 48 | mass cancer metastatic lung tumor patient cell left malignant breast hospital | 21.6 | 25.4 | 12.3 | 30.2 | 18.5 |
| 1 | neo gtt pain resp neuro wean clear plan insulin good | 3.3 | 1.8 | 1.6 | 3.6 | 2.7 |
| 2 | assessment insulin mg/dl plan pain meq/l mmhg chest cabg action | 0.3 | 0.6 | 0.9 | 3.6 | 2.2 |
| 0 | chest reason tube clip left artery s/p pneumothorax cabg pulmonary | 3.2 | 5.5 | 2.5 | 5.6 | 4.0 |
| 25 | c/o pain clear denies oriented sats plan alert stable monitor | 7.3 | 3.9 | 5.9 | 8.2 | 6.5 |
| 47 | pacer pacemaker icd s/p paced rhythm ccu amiodarone cardiac | 8.2 | 9.1 | 8.3 | 13.8 | 10.1 |

Table 2: Top and bottom 5 topics (of 50) based on variance in error rates of groups. Error rates by group and topic $p(\hat{Y} \neq Y | K, A)$ are reported in percentages.

We identified patients with notes corresponding to topic 48, corresponding to cancer, as a subpopulation with large differences in errors between groups. By varying the training size while saving 20% of the data for testing, we estimate that more data would not be beneficial for decreasing error (see Figure 3c). The mean over 50 trials is reported with hyperparameters chosen for each training size. Instead, we recommend collecting more features (e.g. structured data from lab results, more detailed patient history) as a way of improving error for this subpopulation.

Furthermore, we compute the 95% confidence intervals for false positive and false negative rates for a logistic regression with L1 regularization in Figure 3a and Figure 3b.

## 3 Exploring model choice

If a difference in bias is the dominating source of discrimination between groups, changing the class of models under consideration could have a large impact on discrimination.Consider for example Figure 1c in which the true outcome has higher complexity in regions where one protected group is more densely distributed than the other. Increasing model capacity in such cases, or exploring other model classes of similar capacity, may reduce as long as the bias-variance trade-off is beneficial. Bias is not identifiable in general, as this requires estimation or bounding of noise components $N_a$, or an assumption that they are equal, $\overline{N}_0 = \overline{N}_1$, or negligible, $\overline{N}_a \approx 0$. However, as noise is in-dependent of model choice, a difference in bias of different models is identifiable even if the noise is not known,

(a) The false negative rates for logistic regression with L1 regularization do not differ across five ethnic groups, shown by the overlapping 95%-confidence intervals intervals, except for Asian patients.

(b) The false positive rates also does not differ much across groups with many overlapping intervals. Note that Asian patients have high false positive rate but low false negative rates.

(c) Adding training data size on error enrichment for cancer (topic 48) does not necessarily reduce error for all groups. This may suggest we should focus on collecting more features instead.

Figure 3: Additional clinical notes experiments highlight the differences in false positive and false negative rates. We also examine the effect of training size on cancer patients in the dataset.

provided that the variance is estimated. With $\Delta \overline{B} = \overline{B}_0 - \overline{B}_1$, and $\Delta \overline{V} = \overline{V}_0 - \overline{V}_1$, and $\hat{Y}, \hat{Y}'$, two predictors for comparison, we may test the hypothesis $H_0 : \Delta \overline{B}(\hat{Y}) + \Delta \overline{V}(\hat{Y}) = \Delta \overline{B}(\hat{Y}') + \Delta \overline{V}(\hat{Y}')$.

## 4   Regression with homoskedastic noise

By definition of $\overline{N}$, we can state the following result.

**Proposition 1.** *Homoskedastic noise, i.e.* $\forall x \in \mathcal{X}, a \in \mathcal{A} : N(x, a) = N$, *does not contribute to discrimination level* $\overline{\Gamma}$ *under the squared loss* $L(y, y') = (y - y')^2$.

*Proof.* Under the squared loss, $\forall a : \overline{N}_a = \mathbb{E}_X[N(X, a)] = N$, as $c_n(x, a) = 1$. $\square$

In contrast, for the zero-one loss and class-specific variants, the expected noise terms $\overline{N}_a$ do not cancel, as they depend on the factor $c_n(x, a)$.

## 5   Bias-variance decomposition. Proof of Theorem 1.

**Lemma A1** (Squared loss and zero-one loss). *The following claim holds for both:*
*a)* $L(y, y') = [y \neq y']$ *the zero-one loss with* $c_1(x, a) = 2\mathbb{E}[\mathbb{1}[\hat{Y}_D(x, a) = \hat{y}_*(x, a)]] - 1$ *and* $c_2(x, a) = \{1, \text{ if } \hat{y}^*(x, a) = \hat{y}^m(x, a); -1 \text{ otherwise}\}$,
*b) a)* $L(y, y') = (y - y')^2$ *the squared loss with* $c_1(x, a) = c_2(x, a) = 1$.

$$\mathbb{E}[L(Y, \hat{Y}_D) \mid X = x, A = a] = c_1(x, a)\mathbb{E}[L(y, \hat{Y}^*) \mid x, a]$$
$$+ L(\hat{y}^m(x, a), \hat{y}^*(x, a)) + c_2\mathbb{E}[L(\hat{y}^m(x, a), \hat{Y}_D) \mid x, a].$$

*Proof.* See Domingos (2000). $\square$

**Lemma A2** (Class-specific zero-one loss). *With* $L(y, y') = [y \neq y']$ *the zero-one loss, it holds with* $c_1(x, a) = 2\mathbb{E}[\mathbb{1}[\hat{Y}_D(x, a) = \hat{y}_*(x, a)]] - 1$ *and* $c_2(x, a) = \{1, \text{ if } \hat{y}^*(x, a) = \hat{y}^m(x, a); -1 \text{ otherwise}\}$

$$\forall y \in \{0, 1\} : \mathbb{E}[L(y, \hat{Y}_D) \mid X = x, A = a] =$$
$$c_1(x, a)L(y, \hat{Y}^*) + L(\hat{y}^m(x, a), \hat{y}^*(x, a)) + c_2\mathbb{E}[L(\hat{y}^m(x, a), \hat{Y}_D) \mid x, a].$$

*Proof.* We begin by showing that $L(y, \hat{Y}_D(x,a)) = L(\hat{y}^*(x,a), \hat{Y}_D(x,a)) + c_0(x,a)L(y, \hat{y}^*(x,a))$
with $c_0(x,a) = \{+1, \text{ if } \hat{y}^*(x,a) = \hat{Y}_D(x,a); -1, \text{otherwise}\}$.

$$L(y, \hat{Y}_D) - L(\hat{y}^*(x,a), \hat{Y}_D(x,a)) + c_0(x,a)L(y, \hat{y}^*(x,a))$$

$$= \begin{cases} 0, & \text{if } \hat{Y}_D(x,a) = \hat{y}^*(x,a) = 0 \\ -1 - c_0(x,a), & \text{if } \hat{Y}_D(x,a) = 0, \hat{y}^*(x,a) = 1 \\ 0, & \text{if } \hat{Y}_D(x,a) = 1, \hat{y}^*(x,a) = 0 \\ 1 - c_0(x,a), & \text{if } \hat{Y}_D(x,a) = \hat{y}^*(x,a) = 1 \end{cases}$$

As the above should be zero for all options, this implies that $c_0 = 2 * \mathbb{1}[\hat{Y}_D(x,a) = \hat{y}^*(x,a)] - 1$.

We now show that,

$$\mathbb{E}[L(\hat{y}^*(x,a), Y_d) \mid x,a] = L(\hat{y}^*(x,a), \hat{y}^m(x,a)) + c_2(x,a)\mathbb{E}[L(\hat{y}^m(x,a), \hat{Y}) \mid x,a].$$

We have that if $\hat{y}^m(x,a) \neq \hat{y}^*(x,a)$,

$$\begin{aligned}
\mathbb{E}[L(\hat{y}^*(x,a), \hat{Y}_D) \mid x,a] &= p(\hat{y}^*(x,a) \neq \hat{Y}_D \mid x,a) = 1 - p(\hat{y}^*(x,a) = \hat{Y}_D \mid x,a) \\
&= 1 - p(\hat{y}^m(x,a) = \hat{Y}_D \mid x,a) = 1 - \mathbb{E}[L(\hat{y}^m(x,a), \hat{Y}_D) \mid x,a] \\
&= L(\hat{y}^*(x,a), \hat{y}^m(x,a)) - \mathbb{E}[L(\hat{y}^m(x,a), \hat{Y}_D) \mid x,a] \\
&= L(\hat{y}^*(x,a), \hat{y}^m(x,a)) + c_2(x,a)\mathbb{E}[L(\hat{y}^m(x,a), \hat{Y}_D) \mid x,a].
\end{aligned}$$

A similar calculation for the case where $\hat{y}^m(x,a) = \hat{y}^*(x,a)$ yields the claim.

Finally, We have that

$$\begin{aligned}
\mathbb{E}[L(y, \hat{Y}_D)] &= \mathbb{E}[L(\hat{y}^*(x,a), \hat{Y}_D) + c_0(x,a)L(y, \hat{y}^*(x,a)) \mid x,a] \\
&= \mathbb{E}[L(\hat{y}^*(x,a), \hat{Y}_D) \mid x,a] + \mathbb{E}[c_0(x,a) \mid x,a]L(y, \hat{y}^*(x,a)) \\
&= L(\hat{y}^*(x,a), \hat{y}^m(x,a)) + c_2(x,a)\mathbb{E}[L(\hat{y}^m(x,a), \hat{Y}_D) \mid x,a] \\
&\quad + \mathbb{E}[c_0(x,a) \mid x,a]L(y, \hat{y}^*(x,a))
\end{aligned}$$

which gives us our result. $\qquad\square$

Since datasets are drawn independently of the protected attribute $A$,

$$\begin{aligned}
\overline{\gamma}_a(\hat{Y}) &= \mathbb{E}_D[\mathbb{E}_{X,Y}[L(Y, \hat{Y}_D) \mid D, A = a] \mid A = a] \\
&= \mathbb{E}_X[\mathbb{E}_{D,Y}[L(Y, \hat{Y}_D) \mid X, A = a] \mid A = a] \\
&= \mathbb{E}_X[B(\hat{Y}, X, a) + c_2(X,a)V(\hat{Y}, X, a) + c_1(X,a)N(X,a) \mid A = a],
\end{aligned}$$

and an analogous results hold for class-specific losses, Theorem 1 follows from lemmas A1–A2.

# 6 Difference between power law curves

Let $f(x) = ax^{-b} + c$ and $g(x) = dx^{-e} + h$. Then $d(x) = f(x) - g(x)$ has at most 2 local minima.
We see this by re-writing $d(x)$

$$d(x) = ax^{-b} + \tilde{c} - dx^{-e}$$

and so

$$d'(x) = (-b)ax^{-b-1} + dex^{-e-1}$$

Setting the derivative to zero,

$$(-b)ax^{-b-1} + dex^{-e-1} = 0$$

$$x^{b-e} = \frac{ba}{de}$$

which has a unique positive root

$$x = \left(\frac{ba}{de}\right)^{\frac{1}{b-e}}.$$

Since $f(x)$ has a single critical point (for $x > 0$), $f(x)$ can switch signs at most twice. The curves $f(x) = \frac{100}{x^2} + 1$ and $g(x) = \frac{50}{x}$ intersect twice on $x \in [0, \infty]$. If $b = e$, $d(x)$ has a single zero,

$$d(x) = (a - d)x^{-b} + \tilde{c} = 0$$

yields

$$x = (\frac{\tilde{c}}{d - a})^{\frac{1}{-b}} \ .$$