[Reviews · NeurIPS 2018]

Reviewer 1



This paper proposes looking at disparities in classifier outcomes across sensitive attributes as a problem to be remedied by data collection as opposed to classifier modification. The authors advocate decomposing the loss of a classifier into bias, variance, and noise, which allows for a more fine-grained analysis of the sources of disparate outcomes. Based on the this decomposition, the authors show that reducing disparity in outcome may require increasing the bias or variance of predictions. The authors go on to suggest that instead of constraining classifiers to produce parity in outcomes, collecting more data could be sufficient to reduce discrimination. In particular, they make the assumption that the learning curve (error as a function of training set size) follows an inverse power law, an assumption with both empirical and theoretical support. Using subsampling, the authors demonstrate that this curve is a good approximation for the types of errors considered and use the predictions it makes to estimate the reduction in disparity that could be achieved by collecting additional data. They perform several experiments on publicly available datasets to demonstrate these improvements. Finally, the authors suggest a framework for identifying clusters of individuals for whom additional features should be collected. Given a clustering, they propose a metric which indicates that the available features are more predictive on one subgroup than another, meaning that more features are required to improve predictive performance on the disadvantaged subgroup. They perform experiments with simple heuristics to choose clusters, finding some intuitively compelling explanations for disparities in predictive performance. Overall, I think this is a solid paper. It provides a fresh take on a problem that has received a lot of attention lately. Moreover, it provides an approach that, unlike many in the fairness space, could actually be used in practice. Targeted collection of data is far less controversial and more legally feasible than most of the solutions that have been proposed. Technically, the paper makes some nice, clean points; however, to me, the main contribution is conceptual as opposed to technical. The experiments on finding clusters with high disparity seem to be more of a proof of concept than a proposed solution, and I would have liked to see a more structured way of identifying such clusters.

Reviewer 2



The paper focuses on discrimination in machine learning. As opposed to the existing work on discrimination-aware machine learning, the goal of the paper is to actually go beyond the traditional discrimination-accuracy tradeoffs. Instead, the paper proposes a framework to pinpoint the precise sources of discrimination in prediction outcomes, and propose interventions that might enhance both fairness as well as the accuracy of the model. As the paper notes, just constraining the existing (discriminatory) models to remove outcome disparities may not be sufficient. Decreasing accuracy of (one, or) both groups just to achieve equality is definitely an unsatisfactory tradeoff. While previous authors alluded to the possibility of gathering more data to potentially reduce discrimination (https://arxiv.org/abs/1701.08230 and https://arxiv.org/abs/1610.02413), this paper is the first (to the best of the knowledge of this reviewer) to formally tackle this extremely important direction. While the technical contribution is not necessarily huge, the conceptual contribution of the paper would definitely make it a valuable addition to the existing literature on the topic. Detailed comments: - hat{Y} in line 61 is defined as a function of both X and A. However, in Figure 1, both the outcomes as well as predictions are thought to be unaware of A (same as the experimental section). How does one reconcile this difference? Does this have any implications for the analysis that follows? - Figure 1 and Section 3 (until line 113) are very difficult to read. The paper does briefly state that it uses the loss decomposition of (Domingos, 2000), but for a reader not familiar with this framework, it is not entirely clear as to what precisely the Bias and Variance terms defined here are trying to measure. Domingos does provide intuitive explanations for these terms. Perhaps the authors can expand a bit more on these terms, or point the reader to the relevant publication (https://homes.cs.washington.edu/~pedrod/papers/mlc00a.pdf). Fixing this issue can greatly increase the readability of the paper. - It would be nice to see discussion on some other clustering techniques to identify regions where the discrimination between the groups is high. Using uncertainty estimation (e.g., in Gaussian processes) might be one way to do that. - How would the results extend to the statistical parity notion of fairness (https://arxiv.org/abs/1705.09055)? - While not necessary for this submission, it would be great to see some discussion on how the results would extend to the cases of intersectional fairness (https://arxiv.org/abs/1711.05144) -- the cases when one has more fine-grained groups such as gender and race (e.g., African-American men, White women) - Though not necessary for the current submission, it would be interesting to see how the proposed strategy of gathering more data (examples or features) would interact with the delayed-impact-of-fairness framework proposed in the recent study by Liu et al (https://arxiv.org/pdf/1803.04383.pdf). Specifically, is gathering more data on the discriminated group likely to lessen the long term stagnation or decline of these groups?

Reviewer 3



Summary/Contribution This paper shows that disparities in many fairness measures can be decomposed into “bias” (loss caused by deviating from the best feasible classifier), “variance” (loss caused by randomness in the dataset), and “noise” (the loss of the best feasible classifier). In doing so it illuminates some important issues that need to be considered when attempting to improve group fairness measures. For example, many people suspect that high error rates (either FPR or FNR) for a minority group is at least partially due to a lack of minority data. This work shows that these error rates can persist even with plentiful data (ie when the contribution of “variance” to the error rate differences is zero). Crucially, this paper recognizes that “existing approaches for reducing discrimination induced by prediction errors may be unethical or impractical to apply in settings where predictive accuracy is critical, such as in healthcare or criminal justice,” and proposes concrete means to improve algorithms without bearing this unacceptable cost. Weaknesses: The biggest weakness of this paper is its use of the word “discrimination” to refer to differences in cost functions between groups. The authors provide no justification for why differences in (for example) zero-one loss should be termed “discriminatory,” evoking injustice and animus in the reader’s mind. A classifier that tries to predict who will become an NFL quarterback will have a worse zero-one loss for men than women (since no woman has ever played in the NFL), but this is absolutely not discrimination. I’d give this paper an "overall score" of 8 if they used a more value-neutral term (such as “disparities” or “differences”) instead of “discrimination”. The authors also suggest a method for determining where to expend effort to produce better models, which looks at subgroups where error rates differ by protected characteristics. For example, they find that their model seems to make far fewer errors on Hispanic cancer patients than cancer patients of other races. I’d be interested to see whether this effect was due to differences in base rates within the subgroups. One group might be easier to classify simply because their base rate is closer to zero or one, but this doesn’t necessarily suggest the existence of extra features to reduce noise (and therefore the accuracy gap). Clarity: This paper is well written and the math is clear. Originality: This paper is clearly original, although other papers have hinted at similar high-level results. Significance: This paper contains some valuable insights that could be readily applied to existing machine learning workflows, a rare achievement for FATML papers. However, I suspect the practical impact will be small since many problems have lots of data (so bias and variance are small) and no good way to get additional predictive features (so noise can't be reduced). Response to author rebuttal: I was disappointed to see the authors push back about the use of "discrimination" to describe differences in cost functions. "However, in the NFL example, if differences in error rates were related to race, there might be cause for concern--depending on how the classifier is used. Drawing on terminology from prior work, we use “discrimination” as we consider applications in which decisions based on error-biased classifiers may be considered unfair, such as healthcare, employment and criminal justice—similar to how a human making consistent race-dependent errors could also be considered discriminatory." This is all true, differences in error rates *might* be discriminatory, but they also might not be. This is why we need to be extremely precise with our language. Previous papers have been careless with the word "discriminatory", but this does not mean this (otherwise careful and important) paper should follow their example. The title of the paper does not give me confidence that the discussion in the revised paper about when, exactly, differences in cost functions are discriminatory will be sufficiently nuanced.